# The Cholinergic Selectivity of FDA-Approved and Metabolite Compounds Examined with Molecular-Docking-Based Virtual Screening

**DOI:** 10.3390/molecules29102333

**Published:** 2024-05-16

**Authors:** Michael D. Gambardella, Yigui Wang, Jiongdong Pang

**Affiliations:** 1Department of Chemistry and Biochemistry, Southern Connecticut State University, New Haven, CT 06515, USA; 2Department of Chemistry and Chemical & Biochemical Engineering, University of New Haven, West Haven, CT 06516, USA

**Keywords:** cholinesterase, molecular docking, virtual screening, statistical analyses, visualization, molecular dynamics, butyrylcholinesterase, acetylcholinesterase, metabolites, FDA

## Abstract

The search for selective anticholinergic agents stems from varying cholinesterase levels as Alzheimer’s Disease progresses from the mid to late stage. In this computational study, we probed the selectivity of FDA-approved and metabolite compounds against acetylcholinesterase (AChE) and butyrylcholinesterase (BChE) with molecular-docking-based virtual screening. The results were evaluated using locally developed codes for the statistical methods. The docking-predicted selectivity for AChE and BChE was predominantly the consequence of differences in the volume of the active site and the narrower entrance to the bottom of the active site gorge of AChE.

## 1. Introduction

Acetylcholinesterase (AChE) is a hydrolase enzyme found primarily in the postsynaptic neuromuscular junction and responsible for hydrolyzing the neurotransmitter acetylcholine (Ach) into acetic acid and choline [1]. Ach binds to muscarinic receptors on the postsynaptic neuron, which activates an intracellular G-protein-coupled receptor and leads to the dephosphorylation of guanosine triphosphate into its diphosphate form. The resulting cascade from dephosphorylation allows for the influx of K^+^ ions into the neuron [2]. When bound to nicotinic receptors, these receptors undergo a conformational change and are converted into Na^+^ channels at the postsynaptic neuron [3]. The influx of cations leads to depolarization in the neuron, termed the excitatory postsynaptic potential, which increases the probability of a postsynaptic neuron firing off an action potential to release Ach.

Due to AChE’s ability to mitigate Ach hydrolysis and the resultant neuron-to-neuron communication, inhibitors of AChE have found usage as Alzheimer’s Disease (AD) medication. In the later stage of the disease’s progression, however, AChE inhibitors are less effective at improving the well-being of patients. The primary reason for this is a decrease in AChE levels by about 85% in late-stage AD patients [4]. Another cholinergic enzyme, termed butyrylcholinesterase (BChE), has been reported to increase in its concentration in late AD by approximately 120% [5]. BChE hydrolyzes Ach in lieu of AChE, though not as favorably as AChE with respect to its reaction kinetics. Recognition of the marked depletion of AChE and the introduction of higher-than-normal levels of BChE has led researchers to investigate the potential of selective inhibitors of BChE as a treatment option.

To elucidate potential inhibitors of ChEs, molecular-docking-based virtual screening is performed. Molecular docking generates a predefined number of biological conformations, or binding modes, of a ligand in complex with a given protein and calculates the theoretical binding affinity energy between the active site of a protein structure and a ligand. While most studies will choose a ligand dataset to screen against either AChE or BChE, few perform docking against both enzymes and directly compare their binding affinity values.

In the present study, several ad hoc programs were developed to allow for statistical analysis of all the binding modes generated by molecular docking output and the ability to compare individual binding modes with those of a ligand in another enzyme–ligand complex. ZINC ligand datasets were filtered based on select Lipinski parameters and subjected to molecular-docking-based virtual screening using open-source software. The binding affinity values for a given ligand were compared between each ChE.

## 2. Results

Table 1 and Table 2 include compounds with a higher binding affinity from the FDA-approved dataset (separately for AChE and BChE), while Table 3 and Table 4 gather the compounds with a higher affinity from the clean-metabolites-in-vivo dataset. The ligands in these tables were chosen based on their agreeability with Lipinski’s Rule of Five and do not represent the most extreme cases of an energy difference, nor should they be assumed to have appropriate therapeutic windows. The complete and step-by-step results of docking and statistical analyses were grouped together in Figure 1, Figure 2, Figure 3, Figure 4, Figure 5, Figure 6, Figure 7, Figure 8 and Figure 9, Appendix A, and Appendix A. Specifically, the spatial coordinate distributions of heavy atoms excluding carbon from the FDA-approved ligand dataset are shown in Figure 1 (docked against AChE) and Figure 2 (docked against BChE), and those from the clean-metabolites-in-vivo dataset were graphed in Figure 4 (docked against AChE) and Figure 6 (docked against BChE). The statistical occurrence of differences in the binding affinity values between AChE and BChE can be found in Figure 3 for the FDA-approved dataset and in Figure 8 for the clean-metabolites-in-vivo dataset, while a one-graph view of the latter is given in Figure 9. The 10 nearest ChE residues between the ligand cafestrol and the enzymes are illustrated in Figure 5 (AChE) and Figure 7 (BChE). Appendix A shows the typical configuration files for molecular docking, while Appendix A gives pictures of the box sizes (Å) for 1W6R and 4BDS and their relative positions in the proteins and the step-by-step docking results for the representative ligands. Appendix A collects information on selected FDA-approved compounds and their specific interactions with proteins in the first binding mode. Appendix A summarize the complete AutoDock virtual screening results ranked by the differences in the binding affinity (kcal/mol) of the AChE-selective (pp. S37–S89) and BChE-selective (pp. S89–S138) compounds in the FDA-approved ligand dataset and the docking-predicted AChE-selective (pp. S138–S844) and BChE-selective (pp. S844–S1913) compounds in the clean-metabolites-in-vivo ligand dataset, along with the compounds’ respective ZINC ID numbers. Appendix A depicts the “MultiStep” alignment of 1W6R and 4BDS using the VMD program and the “Surf” representation for original ligands of their protein crystals, while Appendix A illustrate the results of molecular dynamic simulations on representative ligand–enzyme complexes.

### 2.1. Molecular Docking

#### 2.1.1. FDA-Approved Dataset—AChE

The hydrogens of the ligands in the FDA-approved dataset were mainly distributed near the AChE residues GLU199, TYR121, and SER122 (behind the dot cluster in Figure 1c). The side chains of these residues provide functional groups to form hydrogen bonds with hydrogen atoms in ligands. For example, glutamic acid (GLU199) has a carboxylic functional group and tyrosine (TYR121) has a phenol group, while serine (SER122) has hydroxymethyl groups in its side chain. To a lesser extent, ligand-bound hydrogens favored the carboxylic side chain of aspartic acid (ASP72) and the imidazole ring in the side chain of histidine (HIS440) for potential intermolecular hydrogen bonding. Both ASP72 and HIS440 can be seen on the cluster border in Figure 1c. The ligand’s oxygen atoms predominantly gravitated towards the hydrogens of SER122, SER200, and TYR121 (all these residues have an OH group in their side chains; they are behind the cluster in Figure 1b). The nitrogen atoms of the ligands in the FDA-dataset targeted the polarized OH groups in the side chains of TYR121 and SER122 and the negatively charged carboxylic group of GLU199, indicating energetically favorable intermolecular hydrogen bonding and coulombic interactions. The oxygens of the ligands were also near the positively charged imidazole side chain of HIS440 (HIS440 is on the cluster border in Figure 1c).

**Figure 1 molecules-29-02333-f001:**
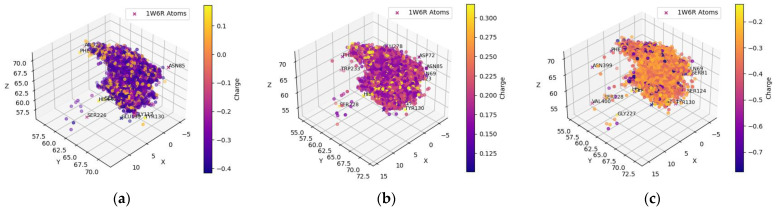
Spatial coordinate distribution of heavy atoms excluding carbon from the FDA-approved ligand dataset docked against AChE. (**a**) Nitrogen spatial coordinates with charge colormap and active site residues within 2.5 Å; (**b**) oxygen spatial coordinates with charge and active site residues within 1.5 Å; (**c**) hydrogen spatial coordinates with charge colormap and active site residues within 1.5 Å.

#### 2.1.2. FDA-Approved Dataset—BChE

The distributions of the atoms were broader in Figure 2 than those in Figure 1. The distribution of the ligands’ hydrogen atoms primarily targeted the OH or COO^−^ groups in the side chains of SER198, GLU197, and TYR128. The hydrogen atoms were further clustered near the BChE residue tryptophan (TRP82), whose side chain had an indole residue, and histidine (HIS438), whose side chain had an imidazole residue. The oxygens of the ligands favored the protons attached to GLY116, TRP82, and THR120. The ligands’ nitrogen atoms similarly favored GLY116 and THR120 but not TRP82. Tryptophan (TRP82) has an indole ring in its side chain, whose size and repulsive nitrogen–nitrogen coulombic interactions may prevent the nitrogen in the ligands from gravitating toward TRP82. The ligand-bound nitrogen atoms were situated near GLU197, and the ligand-bound oxygens were found near HIS438 for possible energy-minimizing electrostatic interactions because the glutamic acid residue was negatively charged and histidine was positively charged in physiological pH environments.

**Figure 2 molecules-29-02333-f002:**
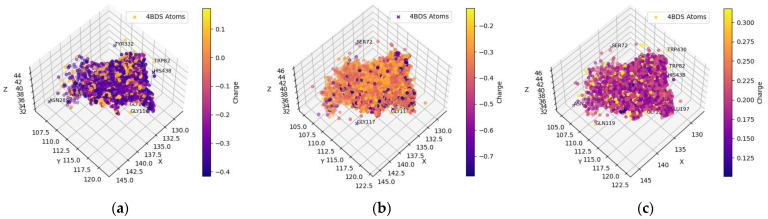
Spatial coordinate distribution of heavy atoms excluding carbon from the FDA-approved ligand dataset docked against BchE. (**a**) Nitrogen spatial coordinates with charge colormap and active site residues within 2.5 Å; (**b**) oxygen spatial coordinates with charge and active site residues within 1.5 Å; (**c**) hydrogen spatial coordinates with charge colormap and active site residues within 1.5 Å.

**Figure 3 molecules-29-02333-f003:**
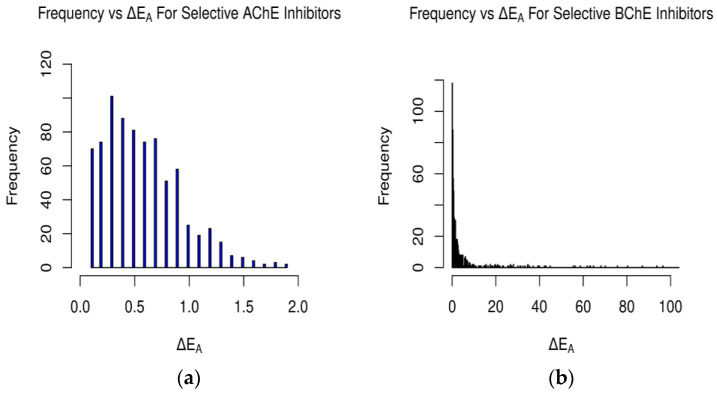
Occurrences of differences in binding affinity values between AChE and BChE for (**a**) selective AChE inhibitors and (**b**) selective BChE inhibitors in FDA-approved dataset. Difference in energy measured in kcal/mol (plot (**b**) cutoff at 100 kcal/mol).

#### 2.1.3. Clean-Metabolites-In-Vivo Dataset—AChE

The distributions of the hydrogen and oxygen atoms were more scattered in Figure 4 than those in Figure 1. Concerning the hydrogen atoms of the ligands found in the clean- metabolites-in-vivo dataset, the most popular oxygen-containing AChE residues were GLU199, TYR70, and TYR121, while the nitrogen primarily targeted by the ligand-bound hydrogen was found in the ASP72 and HIS440 amino acids. Like the FDA-approved dataset, the hydrogens of SER122, SER200, and TYR121 were favored by the oxygen and nitrogen atoms of the ligands. For potential coulombic interactions, the oxygen of the ligands was clustered near the nitrogen in ASP72 at double the frequency of that of HIS440. As with the previous ligand dataset, possible electrostatic interactions with GLU199 were overshadowed by potential intermolecular hydrogen bonding with the neighbor residues TYR121 and SER122.

**Figure 4 molecules-29-02333-f004:**
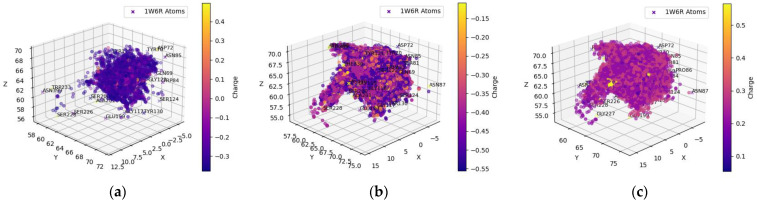
Spatial coordinate distribution of heavy atoms excluding carbon from clean-metabolites-in-vivo ligand dataset docked against AChE. (**a**) Nitrogen spatial coordinates with charge colormap and active site residues within 2.5 Å; (**b**) oxygen spatial coordinates with charge and active site residues within 1.5 Å; (**c**) hydrogen spatial coordinates with charge colormap and active site residues within 1.5 Å.

**Figure 5 molecules-29-02333-f005:**
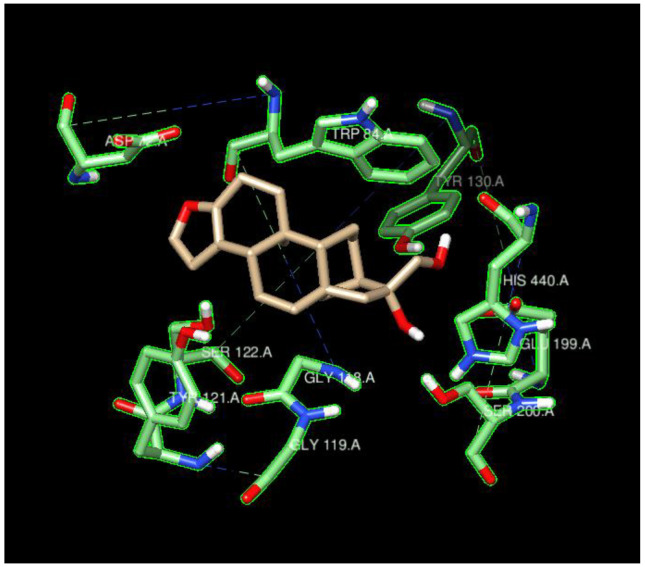
Cafestrol surrounded by the ten AChE residues with highest density of nearby ligand-bound H, O, or N atoms, such as SER200, SER122, TYR121, ASP72, and HIS440.

#### 2.1.4. Clean-Metabolites-In-Vivo Dataset—BChE

As with the FDA-approved dataset, the oxygens of the BChE residues SER198, GLU197, and TYR128 were the primary targets of ligand-bound hydrogen in the clean-metabolites-in-vivo dataset. The hydrogens of the ligands were predominantly clustered near nitrogen-containing HIS438, about twice as frequently as TRP82, following a similar trend to that observed for the FDA dataset’s ligands and BChE residues. The GLY116, THR120, and TRP82 hydrogens were targeted by ligand-bound oxygen, while ligand-bound nitrogen targeted the protons attached to GLY116 and THR120. The negatively charged oxygen of GLU197 and the positively charged nitrogen of HIS438 were the most likely coulombic partners of the nitrogen-bound and oxygen-bound ligands, respectively.

**Figure 6 molecules-29-02333-f006:**
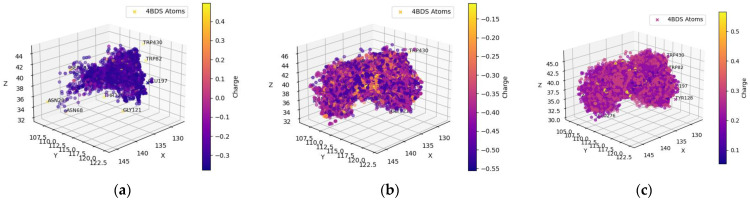
Spatial coordinate distribution of heavy atoms excluding carbon from clean-metabolites-in-vivo ligand dataset docked against BchE. (**a**) Nitrogen spatial coordinates with charge colormap and active site residues within 2.5 Å; (**b**) oxygen spatial coordinates with charge and active site residues within 1.5 Å; (**c**) hydrogen spatial coordinates with charge colormap and active site residues within 1.5 Å.

**Figure 7 molecules-29-02333-f007:**
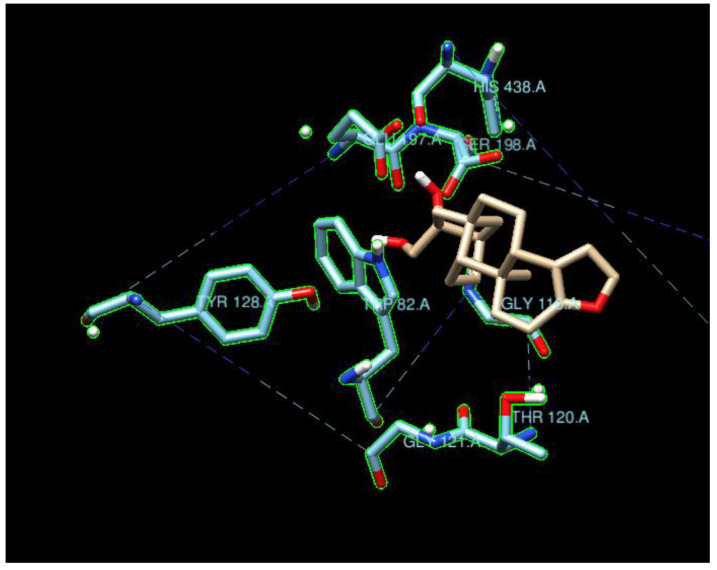
Cafestrol surrounded by the ten BchE residues with highest density of nearby ligand-bound H, O, or N atoms such as SER198, GLU197, GLY116, HIS438, and TYR128. Contrary to Figure 5, the ten nearest residues do not form a significant pocket.

**Figure 8 molecules-29-02333-f008:**
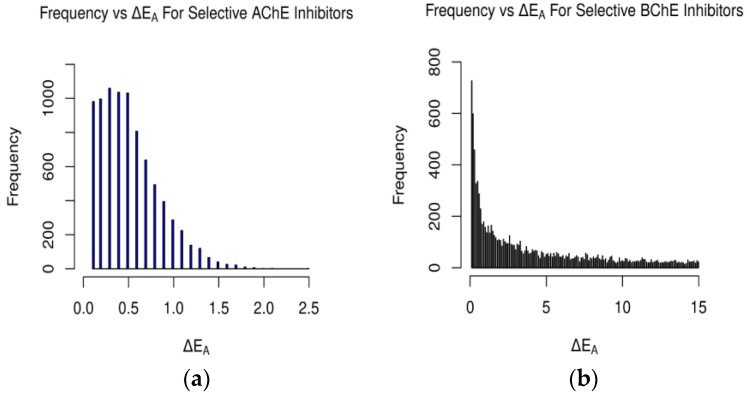
Occurrences of ΔEA values for AChE and BChE for (**a**) selective AChE inhibitors and (**b**) selective BChE inhibitors in clean-metabolites-in-vivo dataset. ΔEA measured in kcal/mol (plot (**b**) cutoff at 15 kcal/mol).

**Figure 9 molecules-29-02333-f009:**
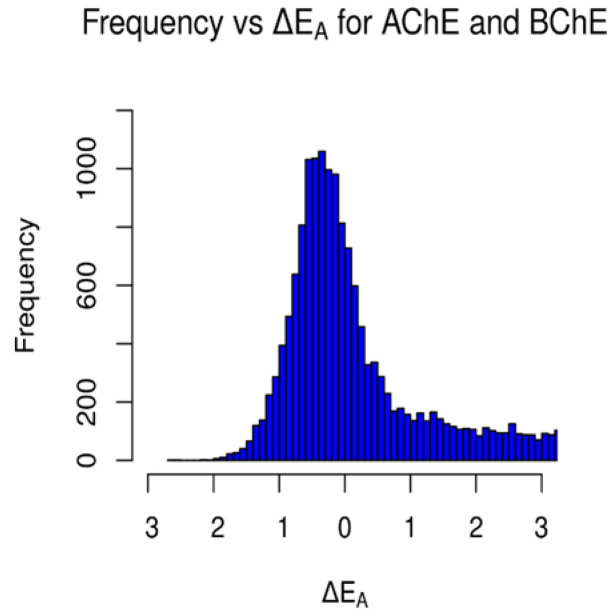
Occurrences of ΔEA values between AChE and BChE for the clean-metabolites-in-vivo dataset. Values to the left of 0 kcal/mol represent the frequency of AChE-selective ligands, and values to the right of 0 kcal/mol represent the frequency of BChE-selective ligands.

The statistical significance of the difference between the FDA-approved dataset’s predicted ΔEA values and the clean-metabolites-in-vivo dataset’s ΔEA values was determined using a nonparametric Mann–Whitney U test. Due to the large sample size of data, the derived U statistic was used to find a Z statistic. From the Z statistic, the *p* value was found to be <<0.001. The results from the test indicate that the difference in the distributions of the two datasets’ calculated ΔEA values is not statistically significant at the >>99.999% confidence interval.

## 3. Discussion

### 3.1. Molecular Docking

The spatial distributions of the O and H atoms of the ligands appeared to be similar, while those of the N atoms were more restricted in all of them (Figure 1, Figure 2, Figure 4 and Figure 6). This is indicative of the fact that there are not as many N atoms in our ligand molecules when compared to O or H. In addition, H atoms uniformly had positive charges and O atoms uniformly had negative charges. While some N atoms depicted in the spatial maps above are positive, many were assigned a negative charge by Open Babel at a pH of 7.4. Therefore, the H, N, and O atoms of the ligands may participate in intermolecular hydrogen bonding and present electrostatic interactions with the residues of the respective proteins. In addition, BChE demonstrated a greater active space volume than AChE according to the spatial distributions (Figure 1 vs. Figure 2; Figure 4 vs. Figure 6).

The typical residues on the proteins that interacted with the ligands also indicated that the main contributions were from hydrogen bonding and coulombic interactions (Figure 5 and Figure 7). The residues either had a charged side chain, such as glutamic acid (GLU), aspartic acid (ASP), and histidine (HIS), or had a polar OH functional group, such as serine (SER) and tyrosine (TYR). It is interesting that glycine (GLY116) appeared in Figure 6, possibly because its small size enables it to interact with ligands using backbone NH and C=O groups. Residues GLY115 and GLY117, along with GLY116, opened up a large space for the ligands to bind.

The ligand cafestrol in Figure 5 and Figure 7 is a diterpenoid molecule present in coffee beans that may be responsible for the biological and pharmacological effects of coffee. The binding affinities of cafestrol with a large search box (30 × 30 × 30) did not indicate there was obvious selectivity for either AChE or BChE (−8.9 kcal/mol and −9.0 kcal/mol) but showed slight BChE selectivity when a small search box (15 × 15 × 15) was used (−7.4 kcal/mol vs. −9.0 kcal/mol). Cafestrol (MW 316.5 g/mol) has a relatively large molecular volume of 503.8 Å^3^. From Figure 5, we see that the ten nearest residues in the AChE active site form a nearly complete pocket, while Figure 7 shows that the ten nearest residues in the BChE gorge do not form a pocket. This and the spatial distribution results highlight the volume difference between the active site gorges of the two ChEs and their potential to bind with auxiliary sites besides the catalytic triad in BChE. The RMSD results from Newtonian dynamics results (Appendix A) suggest that many of the test ligands migrated from their Vina-predicted biological conformation to find a new stabilizing minimum.

For the FDA-approved ligand dataset (Figure 3), approximately 48% (779 ligands) of the molecules were predicted to be AChE-selective (ΔEA > 0.1 kcal/mol) and 46% (739) BChE-selective. The AChE inhibitors had a max energy difference of 1.9 kcal/mol, while the BChE inhibitors’ ΔEA values were over 100 kcal/mol and concentrated from 0.1–10.0 kcal/mol (Figure 3 and Figure 8 and Appendix A). The magnitude of the difference between the docking-predicted binding affinity for AChE and BChE was significant. The data suggest that the larger active site gorge of BChE allows ligands that may not be able to fit into the active site of AChE to bind to it more favorably, resulting in large ΔEA values due to repulsion. The relatively low magnitude of ΔEA for the AChE-selective inhibitors suggests that since the active sites of AChE and BChE both contain the same catalytic triad of residues (SER200, HIS440, and GLU327 in 1W6R), most molecules that can fit into AChE and bind favorably will also bind favorably to BChE, according to Vina and its forks.

A similar gamma-like distribution of occurrences vs. ΔEA was found for the clean-metabolites-in-vivo dataset (Figure 8). About 38% (8376) of the assessed ligands were AChE-selective and 58% (12,745) selective for BChE. The increase in selective BChE inhibitors for the second dataset is a result of the existence of more sterically hindered metabolite compounds than in the FDA-approved set. The predicted AChE-selective inhibitors had ΔEA values that did not exceed 2.7 kcal/mol, while the predicted BChE inhibitors had a wider range of ΔEA values. The concentration of ΔEA values near 0 kcal/mol (Figure 9) is an indication of the similarities between the amino acids of the active sites of AChE and BChE despite their reported 200-cubic-angstroms difference [6].

### 3.2. The Selectivity between AChE and BChE

We then focus on three representative ligands: the AChE-selective ligand ambrisentan (1.9 kcal/mol), the BChE-selective ligand ergotamine (12.6 kcal/mol), and ligand ZINC_253700110, which has a huge affinity difference of ~109 kcal/mol in favor of BChE. These three ligands are all large in the *x*-, *y*-, and *z*-dimensions: ambrisentan, 10.48, 8.38, 7.60 (667 Å^3^); ergotamine, 18.931, 10.122, 6.750 (1293 Å^3^); ZINC_253700110,16.27, 10.90, 9.24 (1638.8 Å^3^). The search box size in docking apparently affected the binding affinities for AChE (Appendix A). When we used a larger box (40 × 40 × 40), the differences in the binding affinities between AChE and BChE were reduced. Ambrisentan possibly was the largest inhibitor that could fit into the bottom of the AChE gorge because large ligands sought to bind other sites of AChE rather than to the bottom of the catalytic gorge (Appendix A) when the binding affinities became positive. This was confirmed by both the results for ergotamine and ZINC_253700110 (Appendix A).

It is possible that ambrisentan has an appropriate conformation to fit into the bottom of the AChE catalytic gorge. Ambrisentan has equal proportions according to its x-, y-, and z-dimensions, while ergotamine and ZINC_253700110 are more extended in one dimension. Appendix A indicates that all three ligands remained in the proximity of the active gorge of 4BDS (ZINC_253700110 remained in the gorge area until a large box of 60 × 60 × 60 was used), while all three ligands started to bind to the open areas on the surface of AChE when the search box reAChEd 40 × 40 × 40. The FDA-approved BChE-selective compounds in Table 2 consistently have larger MW weights (450.93–581.6615 g/mol) and volumes (731.5–1293.4 Å^3^) than the AChE-selective compounds (221.2424–378.5072 g/mol; 238.1–985.5 Å^3^) in Table 1.

Appendix A sequenced 1W6R and 4BDS together. The overall similarity was obvious for the helices and beta-sheets, but there were indeed larger mismatches for the coils that built up the active gorge at the center of the proteins. In addition, the entrance to the bottom of the active gorge was narrower for AChE than for BChE. The residues TYR121 and PHE330 were identified as part of a new sequence to block access to the active gorge in 1W6R [7]. These residues were offset slightly to open up space for two ligands (PFK and an unidentified ligand) in the original 4BDS crystal structure. The narrow entrance was also reflected in the spatial coordinate distributions, in which AChE (Figure 1 and Figure 4) illustrated a thinner expanse than BChE (Figure 2 and Figure 6) near the entrance site.

Caffeine was predicted to be weakly AChE-selective by Vina (Appendix A, −6.7 vs. −6.5 kcal/mol). Caffeine (MW 194.19 g/mol) has a planar structure due to its purine nature. Caffeine’s volume is only 83.8 Å^3^ with four N atoms and two oxygen atoms, so its volume is smaller than that of Tacrine (TAC is the ligand in the gorge of the original 4BDS crystal). Step-by-step docking with different box sizes (Appendix A) indicates that the binding affinity differences are constant as long as the box size is greater than 20 × 20 × 20. An offset of the center of box to the edge increases the difference to 0.8 kcal/mol. However, our preliminary results from kinetic studies showed significant AChE selectivity. It was also reported that caffeine inhibited AChE but not BChE [8]. Three distinct domains were identified as related to the difference in selectivity between AChE and BChE [9]. In addition, 4BDS is heavily glycosylated, with nine N-glycosylation sites (Appendix A).

We can identify caffeine inhibition mechanisms (competitive, non-competitive, or uncompetitive) from kinetic studies. Appendix A shows the changes in the coulombic short-range (Coul-SR) interaction energy and Lennard-Jones short-range (LJ-SR) interaction energy from 50 ns molecular dynamic simulations for caffeine–enzyme complexes. The simulation was conducted using the GROMACS program [10], and further information regarding the molecular dynamics simulation parameters is provided in the Appendix A. Caffeine appears to have a greater combined interaction energy with BChE than with AChE in the 50 ns simulation, so the AChE selectivity might be due to a non-competitive mechanism through caffeine’s binding to the entrance site and its aromatic interactions with TYR121 and PHE330 in 1W6R, which is supported by the stronger LJ-SR component than the Coul-SR energy in general (Appendix A). Further molecular dynamic simulations and experimental kinetic studies were conducted to unveil the AChE-selective mechanism for caffeine and other selected compounds in this paper.

Multiple binding modes were determined by ligand docking (Appendix A). The average binding affinity (<E_A_>) values are usually close to those of the 1st E_A_, and the other statistical parameters do not show significant differences from the 1st E_A_ (Table 1, Table 2, Table 3 and Table 4), which justifies commonly using the first-mode method to compare binding affinities and selectivity. We do see larger differences in $\sigma_{E_{A}}$ for AChE (0.23–1.20) than for BChE (0.13–0.55). Furthermore, the most stable first binding model may also be far away from the active site (Appendix A). Determining selectivity simply using static molecular docking calculations is expected to lead to erroneous predictions.

Both the residue TYR121 and the residue PHE330 at the entrance site appear in all the graphs of the specific interactions between AChE and the selected FDA-approved compounds in Table 1 and Table 2, except for two small ligands (linear Metaxalone and planar Triamterene) (Appendix A). In addition, the exceptional large volume of Doxapram (985.52 Å^3^) does not change its selectivity towards AChE, possibly because of its structural flexibility and equal proportions in three dimensions (12.03, 9.93, and 8.25). On the contrary, Prednisone’s exaggerated length in one dimension (13.11, 5.73, 4.83) and its structural rigidity may limit its selectivity towards AChE, which was confirmed by our additional docking study using a 30 × 30 × 30 box (−8.1 kcal/mol for AChE and −9.0 kcal/mol for BChE in the first binding mode, Appendix A). The molecular dynamics simulation indicates that the van der Waals interaction energies are more significant than the coulombic interaction energies in the cases of caffeine, ergotamine, and ambrisentan. The MD simulation also predicted the correct enzyme selectivity for ergotamine and ambrisentan (Appendix A). In the case of ZINC000253700110, the coulombic interaction energies become dominant over the van der Waals interaction energies (Appendix A). Both the RMSD (Appendix A) and radius of gyration results suggest that the ZINC000253700110-BChE complex underwent more significant adjustments than the ZINC000253700110-AChE complex. Therefore, our results support the lock-and-key model for AChE and the induced fit model for BChE.

## 4. Materials and Methods

### 4.1. Molecular Docking

#### 4.1.1. File Preparation

The structures of AChE (PDB: 1W6R) [11] and BChE (PDB: 4BDS) [12] were obtained from the PD Bank. With MGLTools v. 1.5.6 software, missing or incomplete residues were repaired to ensure an appropriate base structure [13]. The water molecules within the sought-after PDB files were removed due to the binding affinities often being skewed as a result. Protons missing from the original PDB were added to the relevant polar atoms, and Koopmans’ charges were added to all the atoms of the respective enzymes.

An FDA-approved ligand dataset (1614) was acquired from the ZINC15 database [14]. Using the open-source tool Open Babel, the ligand files in the set had hydrogen atoms added to suit a biological pH of 7.4, partial charges implemented, and an MMFF94 molecular forcefield applied [15]. Another ligand dataset of 37,091 molecular files with the ZINC15 classifiers metabolites (primary and secondary), in vivo (ligands in the dataset were tested in vivo), and clean (no thiol or phosphate groups) was downloaded from the ZINC15 database. Ligands with >18 rotatable bonds were filtered using a Python script. The remaining ligands were subject to the same preparation as the FDA-approved dataset.

#### 4.1.2. Docking Parameters

The open-source software AutoDock Vina 1.2.0 and its fork, QVina2.0, was employed to carry out the molecular docking [16]. To perform docking-based virtual screening, a Perl script was written to automatically dock individual ligands within the datasets. The FDA-approved dataset was docked against the prepared ChEs using Vina at the San Diego Supercomputer Center (SDSC). The filtered ligands from the clean-metabolites-in-vivo dataset underwent docking against both ChEs. Both datasets’ ligands were docked under varying binding modes (6–20) and exhaustiveness value parameters (5–20).

### 4.2. Docking Analysis

#### 4.2.1. Binding Affinity

A program was written in C++ to read in and store the ligand names, the numbers of modes generated, the orders of the binding modes, the binding affinities, and the lower bound/upper bound RMSD (root mean square deviation) scores. The mean, standard deviation, and standard error of the binding affinities of each binding mode per ligand were calculated and stored into respective vectors. The difference between the AChE and BChE binding affinities of the lowest-energy binding mode was calculated and tabulated per ligand. A sorting algorithm was implemented to filter ligands by the calculated parameters and print them to an output file. An R script was written to read the generated output file and produce plots of the differences in the binding affinity per ligand per enzyme using the ggplot2 library.

#### 4.2.2. Residues of Interest

A series of ad hoc Python scripts were developed to assess which amino acids of the active site were targeted the most frequently by the atoms of the docked ligands. The spatial coordinates and charges of the lowest-energy binding mode of the ligands were parsed from their respective .pdbqt files. A distance calculation was performed between the atoms of the ChE residues and the atoms of the ligands to determine which amino acid ligands in the dataset were clustered near to the most frequently. The atom types of the ligands and residues could be compared individually. Plots of the distribution of ligands were generated utilizing the matplotlib library.

## 5. Conclusions

While molecular-docking-based virtual screening enables us to dock and calculate the theoretical binding affinity of large libraries of ligands against enzymes of a similar tertiary structure, these calculations are inherently limited in their ability to accurately predict the enzyme selectivity in vitro/in vivo of a ligand for enzymes of a similar structure. The statistical analyses above may provide pictures of the docking-predicted enzyme selectivity trends from a set of FDA-approved and metabolite compounds. The observed docking-predicted selectivity between AChE and BChE may be related to the following reasons: first, AChE has larger space at the bottom of its active gorge (near the catalytic triad); second, AChE has a narrower entrance for ligands to feasibly enter the active site when compared to the entrance of BChE; finally, the active site of AChE’s structure is relatively constrained/inflexible when compared to that of BChE. Furthermore, the larger active space of BChE tends to accommodate relatively larger ligands in the gorge area, while AChE must bind large ligands on sites farther away from the active gorge.

## Figures and Tables

**Table 1 molecules-29-02333-t001:** Results from ligands with higher binding affinity for AChE in FDA-approved dataset (10 modes) *.

Ligand	ΔE_A_	1st E_A_	<E_A_>	$\sigma_{E_{A}}$	<rmsd l.b.>	<rmsd u.b.>
Ambrisentan	1.9	−10.8	−9.0	0.97	1.855	3.895
Methadone	1.9	−9.9	−8.9	0.60	1.696	4.326
Triamterene	1.8	−9.6	−8.4	0.45	2.176	4.693
Prednisone	1.7	−11.0	−8.7	1.20	1.639	4.357
Metaxalone	1.6	−8.8	−8.0	0.59	2.085	3.697
Doxapram	1.5	−10.9	−9.2	1.07	1.617	3.620

* ΔE_A_—the difference in energy between AChE and BChE; 1st E_A_—lowest energy binding mode of AChE–ligand complex; <E_A_>—average of all binding modes’ energy values for AChE–ligand complex; $\sigma_{E_{A}}$—standard deviation of binding modes energies of AChE–ligand complex; <rmsd l.b.> and <rmsd u.b.>—the mean of all binding modes’ root mean squared deviation values for AChE–ligand complex (lower bound and upper bound). All ΔE_A_ values in kcal/mol and RMSD values in angstroms.

**Table 2 molecules-29-02333-t002:** Results for ligands with stronger binding affinity for BChE in FDA-approved dataset (10 modes).

Ligand	ΔE_A_	1st E_A_	<E_A_>	$\sigma_{E_{A}}$	<rmsd l.b.>	<rmsd u.b.>
Ergotamine	12.6	−12.0	−11.5	0.31	2.568	8.325
Ciclesonide	9.9	−11.8	−11.0	0.49	2.789	5.359
Suvorexant	7.0	−10.1	−9.1	0.30	4.740	8.674
Nintedanib	6.8	−10.6	−10.2	0.24	2.664	6.214
Lurasidone	6.0	−11.1	−11.1	0.55	2.985	5.905
Amcinonide	5.0	−10.9	−10.2	0.40	3.320	6.541

**Table 3 molecules-29-02333-t003:** Results for ligands with stronger binding affinity for AChE in the clean-metabolites-in-vivo dataset (6 modes).

Ligand	ΔE_A_	1st E_A_	<E_A_>	$\sigma_{E_{B}}$	<rmsd l.b.>	<rmsd u.b.>
Afzelchin	2.1	−10.2	−8.3	0.95	1.937	6.013
Aminoclonazepam	1.9	−10.7	−9.3	0.78	2.708	4.454
Dihydroisorhamnetin	1.7	−10.2	−9.0	0.68	1.254	5.850
Bisphenol A	1.5	−9.3	−8.7	0.42	1.989	5.145
N-cinnamoyloctopamin	1.3	−10.4	−10.0	0.24	2.816	4.520

**Table 4 molecules-29-02333-t004:** Results from ligands with stronger binding affinity for BChE in the clean-metabolites-in-vivo dataset (6 modes).

Ligand	ΔE_A_	1st E_A_	<E_A_>	$\sigma_{E_{A}}$	<rmsd l.b.>	<rmsd u.b.>
Azukisapogenol	15.0	−10.5	−10.1	0.23	1.926	2.790
D-maslinic acid	11.0	−10.6	−10.2	0.25	1.687	4.801
Isoliensinine	7.4	−11.3	−10.9	0.26	2.491	6.986
Dukunolide D	4.8	−11.0	−10.8	0.13	2.211	5.002
Zanthobisquinolone	3.2	−11.1	−10.7	0.23	1.770	5.954

## Data Availability

Data are contained within the article and Appendix A.

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
