# Peer review of "The Cholinergic Selectivity of FDA-Approved and Metabolite Compounds Examined with Molecular-Docking-Based Virtual Screening"

_molecules, 2024, doi:10.3390/molecules29102333_

Round 1
Reviewer 1 Report
Comments and Suggestions for Authors
The article titled " Cholinergic Selectivity of FDA-approved and Metabolite Compounds Examined with Molecular Docking-based Virtual Screening" performed docking-based virtual screening for searching selective modulators targeting AChE and BChE. The work could be acceptable for this journal after major revision. The following are the questions in this manuscript:
Comment 1: The necessary binding models between FDA-approved modulators and AChE/BChE need to be thoroughly discussed, particularly focusing on the crucial interactions of ligands and receptors.
Comment 2: Please provide the experimental data of those FDA compounds targeting AChE and BChE in order to facilitate a better comparison of affinity and selectivity through virtual screening.
Reviewer 2 Report
Comments and Suggestions for Authors
Authors presented important theoretical research on acetylcholinesterase and butyrylcholinesterase inhibitors, results are valuable for design drugs for application as Alzheimer’s Disease medications. Selectivity issues are discussed, difference between AChE and BChE active sites are established, the best ligands were revealed. The manuscript can be published after minor revision. Some comments:
1) Line 122 - serine contains hydroxymethyl group (not ethyl hydroxyl groups
2) Table 1. It would be better to put only the title above the table, and below the table: ΔEA - the difference in energy between AChE and BChE, etc
3) Lines 150-151 – “residue” is more correct than “group” for indole and imidazole.
4) References to some figures and tables are absent in the Chapter 3.
5) Line 336. “Due to purine nature” is better than “adenosine-related structure” about caffein.
6) Line 347 – uncompetitive
7) Please format references properly (Author 1, A.B.; Author 2, C.D. Title of the article. Abbreviated Journal Name Year, Volume, page range).
Round 2
Reviewer 1 Report
Comments and Suggestions for Authors
The revised manuscript resolved my concerns about binding models between FDA-approved modulators and AChE/BChE, and the authors also deeply disscussed the interactions of ligands and recepters. I suggest this manuscript can be accepted.
